# Mycotoxin Analysis of Grain via Dust Sampling: Review, Recent Advances and the Way Forward: The Contribution of the MycoKey Project

**DOI:** 10.3390/toxins14060381

**Published:** 2022-05-31

**Authors:** Biancamaria Ciasca, Sarah De Saeger, Marthe De Boevre, Mareike Reichel, Michelangelo Pascale, Antonio F. Logrieco, Veronica M. T. Lattanzio

**Affiliations:** 1Institute of Sciences of Food Production (ISPA), National Research Council of Italy (CNR), Via Amendola, 122/O, 70126 Bari, Italy; antonio.logrieco@ispa.cnr.it (A.F.L.); veronica.lattanzio@ispa.cnr.it (V.M.T.L.); 2Centre of Excellence in Mycotoxicology and Public Health, Faculty of Pharmaceutical Sciences, Ghent University, Ottergemsesteenweg 460, B-9000 Ghent, Belgium; sarah.desaeger@ugent.be (S.D.S.); marthe.deboevre@ugent.be (M.D.B.); 3Eurofins WEJ Contaminants, Neuländer Kamp, 1, D-21079 Hamburg, Germany; mareikereichel@eurofins.de; 4Institute of Food Sciences (ISA), National Research Council of Italy (CNR), Via Roma, 64, 83100 Avellino, Italy; direttore@isa.cnr.it

**Keywords:** dust, wheat, mycotoxins, sampling, LC-MS/MS, EC Regulation

## Abstract

The sampling protocols for the official control of the levels of mycotoxins in foodstuffs are very costly and time-consuming. More efforts are needed to implement alternative sampling plans able to support official control, or to adapt the current ones. The aim of the research carried out within the European Horizon 2020 MycoKey project was to evaluate the applicability at industrial scale of the dust sampling approach to detect multiple mycotoxins in grains. To this end, two trials were performed on an EU industrial site: (i) control of the unloading of wheat from train wagons; (ii) control of the unloading of wheat from trucks. In line with previous studies, the MycoKey results indicated that dust sampling and mycotoxin analysis represent a fitness for purpose approach for non–destructive and rapid identification of wheat commodities compliant to the maximum permitted levels. Based on reviewed and newly generated results, this article discusses potential applications and limits of the dust sampling methodology, identifying future research needs.

## 1. Introduction

The sampling and analysis (SA) plans used for official mycotoxin control vary in different regions of the world. At the global level, the Codex Alimentarius Commission on Food, the joint intergovernmental body of the Food and Agriculture Organization (FAO) and the World Health Organization (WHO), is the main standard-setter. Codex Stan CXS 193-1995 and its amendments define the sampling plans and performance criteria for analytical methods for aflatoxins in peanuts, maize and tree nuts [1]. These standards can be applied voluntarily by Codex members, and the national mycotoxin legislation in many countries is based on them. In the European Union (EU), the methods of sampling and analysis for the official control of the levels of mycotoxins in foodstuffs are described in the Commission Regulation (EC) No. 401/2006, as amended by Commission Regulations (EC) No. 178/2010 and EC No. 519/2014, covering methods of sampling and analysis of mycotoxins in foodstuffs [2,3,4]. Furthermore, the International Organization for Standardization (ISO) standard, ISO 24333:2009, provides the requirements for dynamic or static sampling of cereals and cereal products by manual or mechanical means, although they are not binding [5]. In the USA, guidance about all aspects regarding regulatory operations, such as sample collection and sample analysis for the main mycotoxins in major food commodities are provided in the Compliance Guidance Program Manual [6].

The official sampling protocols are very challenging in practical terms. Some of the practical difficulties encountered in the application of the official European sampling plan were summarized by Spanjer [7]. For example, in the import control of tree nuts packed in sacks, two food inspectors would need half a working day to sample only one container on just one ship. Regarding the sampling method for cereals, one major issue was the collection of a representative sample from a large lot of cereals and cereal products (lots ≥ 50 tons). In this context, the Commission Regulation (EC) No. 401/2006 states: “If it is not possible to carry out the method of sampling described above because of the unacceptable commercial consequences resulting from damage to the lot (because of packaging forms, means of transport, etc.) an alternative method of sampling may be applied provided that it is as representative as possible and is fully described and documented” [2]. However, the Directive lacked guidance about what is considered “as representative as possible”. The subsequent Regulation EC No. 519/2014 [4] provided guidance for sampling for large lots, or lots stored or transported in a way in which sampling throughout the lot is not feasible. The Regulation introduces the possibility of sampling only a smaller part of the whole lot (at least 10% for food commodities) for the estimation of the whole lot. However, in case of dispute, if the sampling of part of the lot was questioned by the food business operator, the Competent Authority will be allowed to sample throughout the whole lot, at the owner’s cost. Therefore, the Regulation determines an increase in costs for companies, and does not consider that the main barrier to official sampling concerns the budget and resources. While SA plans that are compliant with the regulatory framework should be used for official controls, food and feed business operators could apply an alternative sampling and analysis approach. More work is needed in this area, to implement cost and time-effective SA plans able to support official control or to update current ones.

The aim of the present manuscript is to review the available studies in this field, present new results generated in the MycoKey Project, discuss the potential applications and limits of the dust sampling methodology and identify future research needs.

## 2. Results

### 2.1. Review of Available Studies

A first study was carried out by Focker et al. [8], aimed to identify the most cost-effective SA plan for deoxynivalenol (DON) in wheat, and aflatoxin B1 (AFB1) in maize in truckloads’ or ships’ compartments of cereals at arrival. Considering that, for a predefined budget, different analytical methods, such as instrumental methods (e.g., liquid chromatography coupled to mass spectrometry (LC-MS/MS)), enzyme-linked immunoassay (ELISA) or lateral flow device (LFD) allow the analysis of a different number of samples, the authors developed an optimization model considering as decision variables the choice of the analytical method, the number of incremental samples and the number of aliquots analyzed. The model resulted in an optimized sampling plan for each analytical method, where the number of incremental samples collected and the number of analyzed aliquots were optimized, in order to maximize the number of correct decisions on accepting/rejecting batches within a predefined budget. Specifically, for DON in a wheat batch of 100 tons, the most cost-effective SA plan was based on LFD analysis performed on site directly on the incremental samples (i.e., not combined in the aggregated sample), allowing 89.7% of correct decisions within a budget of 500€. For budgets above 800€, the difference in estimated percentage of correct decisions between optimal SA plans based on laboratory analysis by ELISA or LC-MS/MS was minimal. Different conclusions were obtained for AFB1 in a maize batch of 100 tons. AFB1 in maize is a more heterogeneously distributed mycotoxin and more incremental samples must be collected and combined in an aggregate sample to obtain a representative contamination value. Therefore, the most cost-effective solution for AFB1 in maize involved SA using either ELISA or LC-MS/MS, where the incremental samples were combined into an aggregate sample and only a few aliquots were analyzed, while the SA plan using LFD resulted in being more expensive, since each incremental sample must be analyzed individually. 

Besides practical aspects, another limitation of the current Regulations on mycotoxin sampling is that the general principles, schemes and sampling plans adopted so far are mainly based on those for aflatoxins, neglecting the different variability of mycotoxins in food products under specific conditions of agroecological production [9]. For example, in the Directive EC No. 401/2006 [2], and subsequent amendments (EC No. 178/2010 [3] and EC No. 519/2014 [4]), the same sampling procedure is provided for the analysis of aflatoxins and *Fusarium* toxins in cereals, and it is defined according to cereal batch weight. However, the distribution of *Fusarium* toxins is much more homogeneous than the aflatoxin distribution [10,11]. This implies fewer samples should be collected to obtain a representative result in the case of *Fusarium* toxins, with respect to aflatoxins. Since representative sampling for aflatoxins is more difficult than sampling for other known mycotoxins in foodstuffs, the recommended sampling procedures for aflatoxins are adapted to other mycotoxins [12]. However, this approach does not consider the practicalities and costs involved in collecting a greater number of samples. To define appropriate sampling plans, knowledge of the distribution of contaminated units within the bulk is essential. 

To date, the variability and the distribution were studied for twenty-six different mycotoxin/commodity combinations [13,14,15,16,17,18,19,20,21,22,23,24,25,26,27,28,29,30,31,32,33,34,35,36,37,38,39,40,41]. In Table 1, the distribution study for each mycotoxin in several commodities is indicated by a symbol (●). As outlined in Table 1, the focus of most published studies is on aflatoxins [13,14,15,16,17,18,19,20,21,22,23,24,25,26,27,28,29,30,31,32], whereas studies concerning sampling plans for DON are centered on cereals (wheat [33], maize [10] and barley [34]). In particular, for cereals, sampling plans for maize were defined for aflatoxins [11,22,23,24], fumonisins (FBs) [35,36,37], DON [10] and ochratoxin A (OTA) [38]. Data are also available for: DON in wheat [33]; DON in barley [34] and OTA in wheat and oats [38]. 

A detailed analysis of the sampling plan performance for all of the mycotoxin/commodity combinations can be performed by the Mycotoxin Sampling Tool (released by FAO in 2014 and available at www.fstools.org/mycotoxins, accessed on 8 May 2022). 

An alternative strategy for detecting mycotoxins and overcoming the problem of their variability in different commodities could be provided by the indirect sampling of grain dust. Dust is generated by particle abrasion or friction any time the grain is moved or transferred, whether it be by chute, bucket elevator, transfer belt or open fall [42]. Dust also accumulates in mills and storage facilities. The amount of grain dust generated will depend on the degree of attrition involved in the operation and type of grain handled. Oats and rye, for example, generally generate more dust in the same operation than do wheat or maize [42]. The mycotoxin distribution in the distinct grain tissue is related primarily to the colonization pattern of the mold. However, mycotoxin distribution could also depend on the level and time of fungal infection, to the mycotoxin structure (hydrophilic/hydrophobic) and to the pre- and postharvest conditions [43]. One of the advantages of sampling dust particles instead of a smaller part of the whole lots was that the former should be more representative of the entire batch, as they result from a large number of grains [44]. A prerequisite for indirect measurements is that the concentration of mycotoxins in the lot is related to the concentration measured in the dust [45]. Furthermore, a reliable statistical model that explains the relationship between the mycotoxin level in the lot and the dust sample collected from it should be implemented. Different dust sampler designs were tested so far.

The starting point of dust sampling is the trials described by Stroka et al. [46] for analysis of mycotoxins in the dust of food bulk ware. Dust particles smaller than 2500 µm were collected from samples on different filter materials by means of a sampling probe or the DiscoveryCERT FQS™ sampling system. The use of filters to collect the dust had the following disadvantages: only a single analysis could be carried out by extracting the entire filter and it was not possible to avoid the collection of fine dust. The latter is ubiquitous in grain facilities and cannot be used to determine the contamination of a specific lot. The study showed neither a statistical model explaining the relationship between the sampled bulk material and dust collected from it, nor a Pearson correlation coefficient for measuring the relational resistance between the two variables. However, from the scatter plot of the OTA concentration in dust (OTADust) versus OTA concentration in green coffee beans of the whole material (OTABulkMaterial), an OTADust/OTABulk Material ratio of approximately 20 was calculated. The ratio was based on seven sampled batches of about 3–10 kg collected by the sampling probe.

In 2010, Eurofins developed a patented procedure based on a filter-free, cyclone-based sampler and implemented a statistical model to correlate mycotoxin contamination in dust with food bulk material [47]. Based on this, the technology was further improved resulting in the development of the rapidust^®^ system, a mobile dust sampler, and the first proof of concept studies for mycotoxin grain control [44]. The rapidust^®^ system consisted of a 1.5 m stainless steel sampling lance connected to a cyclone that was attached to a suction unit. A probe head on the lance was protected by a grid, in order to exclude large particles (>2000 µm) from entering the system. Particles between 100–1000 µm were collected in jars attached to the cyclone by an adaptor from 78 wheat batches (1–10 kg) and 52 rye batches (1–10 kg), coming from several European countries and with different harvest years, quality (feed and food) and processing grade. A ratio of 40 and 7.5 between the mycotoxin concentration in the wheat samples and dust sample was calculated for DON and ZEN, respectively, whereas a ratio of 7.3 between the DON concentration in the rye samples and dust samples was found. The integration of 78 data points for wheat samples resulted in a linear correlation model with a correlation coefficient (R^2^) of 0.85 and 0.82 for DON and ZEN, respectively. A linear model was also found for DON in 52 rye samples (R^2^ = 0.73). 

In parallel, studies were carried out on the sampling of wheat dust with a laboratory dust collection system and subsequent analysis of DON by LC-MS/MS or ELISA within the European FP7 MycoHunt project (“Rapid biosensor for the detection of mycotoxin in wheat”) [45,48,49]. The dust was produced from the wheat samples by the use of a dust collection facility. Specifically, wheat samples were transferred through the dumping pit to the vertical elevator system via the Archimedes screw. The dust collection device (i.e., a vacuum cleaner) was mounted at the highest point of the system, where the grains fall into the vertical bin. The dust was collected by removing the paper bag from the vacuum cleaner. In these studies, dust sample particles <50 µm were collected by sieving light fractions (<2500 µm). A linear correlation between the DON content in wheat and in corresponding dust was found either in a set of 12 wheat samples analyzed by LC-MS/MS (R^2^ = 0.94) [45], or in a set of 16 wheat samples (R^2^ = 0.89) analyzed by ELISA [48]. The ratio between DON contamination in dust and wheat was evaluated by the value of slopes of linear regression (13 and 5, respectively). In another study [49], 40 wheat samples (5 wheat varieties × 2 agronomic practice × 4 replications) highly contaminated by DON (range between 1450 µg/kg and 10,670 µg/kg) and the corresponding dust samples were analyzed. A sigmoidal relationship (y = 5.9 ln(x) − 28.4) was found. By comparative analysis of the results of the different studies [45,48,49] the authors suggested that the relationship depended on the concentration range of DON in wheat. In particular, the wheat samples at DON concentrations lower than 1250 µg/kg were fitted by a linear model, while a sigmoidal model was more suitable for the samples contaminated at DON concentration higher than 1250 µg/kg. Furthermore, the sigmoidal model showed a 5.9 fold accumulation in dust with respect to wheat for DON.

Recently, Limay-Rios [50] conducted a study to detect 21 mycotoxins, including DON and its acetylated forms, deoxynivalenol-3-glucoside, T-2 toxin, fumonisins, moniliformin, zearalenone, beauvericin, enniatins, *Alternaria* metabolites, citrinin and ochratoxins in winter wheat by aspirating wheat dust particles using a drum vacuum cleaner (RIDGID WD14500; 8.0 kW and 53 L; Ridge Tool Company, 400 Clark St, 44035, Elyria, OH, United State) equipped with an in-line air filter (FleetguardOptiAir 1100 series; Cummins Filtration, Nashville, TN, USA). The system was designed to collect only particles with a size <1650 µm. A total of 323 wheat samples were evaluated in this study, of which 182 field samples (5–10 kg) collected downwind as the combine was unloading into a truck during harvest, 92 storage samples (5–10 kg) naturally contaminated were collected from on farm bins and 49 storage samples (6 kg) were collected from small-scale storage bins seeded with an OTA producer, *Penicillium verrucosum*. The storage samples were collected at the unloading in the hopper. Among the targeted mycotoxins, DON, 15-acetyl DON, T-2 toxin, zearalenone, enniatins, ochratoxins citrinin, penitrem A, alternariol showed a strong linear relationship between the mycotoxin content in grain and that in grain dust (R^2^ > 0.65). Specifically, the best correlation coefficients were found for OTA and DON (= 0.95). Ratios of mean concentration for each mycotoxin in dust vs. grain varied greatly among several mycotoxins, with values ranging from 62 to 4. The accuracy of the proposed model (linear regression) was evaluated based on the root-mean-square error (RMSE) at the recommended maximum limits (MLs). The RSME measures the differences between values predicted by the model and the corresponding observed values, while the interval of 2 RSME was used to evaluate the model accuracy (i.e., to verify that observations fall in the range predicted by the model with a probability of 95%). An accurate model to estimate the mycotoxin concentration in uncleaned grain was only implemented for DON. The RMSE estimated of 293 µg/kg (interval ± 586 µg/kg) is suitable for determining the mycotoxins’ concentration in grain samples contaminated at 2000 µg/kg (Health Canada’s proposed ML in uncleaned wheat), but is not satisfactory to predict sample contamination at 1000 µg/kg (Health Canada’s proposed ML in uncleaned wheat for use in baby food). In fact, the error percentage (predicted value/2RMSE × 100) was 30% in the first case (ML 2000 µg/kg) and 59% in the second one (ML 1000 µg/kg). Based on the estimated RMSE values, the model resulted in being not adequate for the other targeted mycotoxins. It is worth mentioning that, in the case of DON, the less restrictive ML, but also its high detection frequency and its high concentration compared to the other mycotoxins, were determining factors in implementing an accurate and reliable model.

### 2.2. MycoKey Results

The aim of research carried out within the European Horizon 2020 MycoKey project was to evaluate the applicability at industrial scale level of the dust sampling approach to detect multiple mycotoxins in grains. The strategy implemented in the MycoKey project involves the application of the rapidust^®^ system, the mobile dust sampler developed by Eurofins [44]. This sampler system allows the collection of only the particle size fractions between 100–1000 µm by means of a vacuum stream and a cyclone type collector, as described by Reichel et al., 2014 [44].

To this end, two trials were performed in an EU industrial site, control of wheat unloading from train wagons (trial 1) and the control of wheat unloading from trucks (trial 2). As shown in Figure 1, the procedure consisted of four main steps: 1. Sampling of grain and dust; 2. analysis by a confirmatory method (LC-MS/MS); 3. Set-up of a correlation model; 4. Verification of the model. In step 3, the sample set 1 collected from trial 1 and the sample set 2A from trial 2 were used to set up a model to correlate the content of the main mycotoxins in kernels, and in the respective dust sample. In step 4, the implemented model was applied to an additional set of dust samples and grain samples collected from trial 2 (sample set 2B), and the results obtained in the whole grain using the implemented model were compared with the concentration of mycotoxins directly detected in the whole grain lot to verify the model.

#### 2.2.1. Dust Sampling 

Two trials were performed on an industrial scale to test the implementation of dust sampling procedures for the grain industry. The first trial (trial 1) was focused on the control of wheat upon delivery by a train, while the second one (trial 2) was focused on the control of wheat upon delivery by trucks (Figure 2). Six wheat and dust samples were collected from each train wagon in trial 1 (sample set 1), whereas two sample sets were obtained in the trial 2: a sample set of 32 wheat and dust samples collected from each of the trucks (sample set 2A), and an additional set of 8 dust samples collected by randomly selecting 8 of the 32 wagons (sample set 2B). 

#### 2.2.2. Multi-Mycotoxin Analysis in Grain and Dust by LC-MS/MS

Grain and dust collected by the rapidust^®^ system were analyzed by a LC-MS/MS method to detect the contamination level of NIV, DON, T2, HT2, ZEN, FB1, FB2 and OTA, after verification of in-house analytical performances. A preliminary estimation of the uncertainty was calculated using the Horwitz equation, as suggested by the Codex Alimentarius Commission Guidelines [51]. More accurate approaches based on interlaboratory validation, proficiency test data and data from the analysis of reference materials were not applicable because of the absence of these data for dust samples at the time of the study (main MS parameters, the result of in house verification performances and the uncertainty for grain and dust samples are provided in Appendix A, respectively, of the Appendix A). The results of the multi-mycotoxin analysis in wheat grains and dust from the two trials are reported in Table 2. 

In both trials, the contamination levels in grain samples for most of the mycotoxins analyzed were lower than LOQ. Grain samples were only contaminated with DON and the sum of T-2 and HT-2 with an incidence of 41% and 38%, respectively. Otherwise, a higher incidence of mycotoxins was found in the corresponding dust samples. In the first trial, all of the dust samples were contaminated with NIV, DON, T-2 and HT-2, whereas in the second trial only DON was detected in all of the dust samples. An incidence of 97% was found for NIV and the sum of T-2 and HT-2, otherwise ZEN and OTA were detected in 16, and 38%, of the sampled dust, respectively. Co-occurrence of type-A and -B trichothecenes, ZEN and OTA confirmed the fitness for purpose of multi-mycotoxin detection methods to control the incoming materials. DON was found at higher concentrations and more frequently than any other mycotoxin in both wheat and respective dust, with concentration averages of 38 µg/kg in grain (DONgrain) and 3286 µg/kg in dust (DONdust). 

The average ratio DONdust/DONgrain was 123 ± 43 for DON (*n* = 15, only positive samples of dust and grain were considered, particle size (ps): 100–1000 µm). This study showed a much higher dust-to-grain ratio for DON than reported in other published studies. A DONdust/DONgrain ratio of 13.1 (*n* = 12, dust particle size (p.s.) <50 μm) was found in wheat samples by Sanders, 2013 [13], whereas a value of 5.9 and 5.4 was found by Reichel, 2014 [44] (*n* = 78, ps: 100–1000 µm) and Limay-Ros [50] (*n* = 323, ps < 1650 µm), respectively. Higher levels of mycotoxins in dust than in grain may be due to the distribution of *Fusarium* mycotoxins in infected cereals [52,53,54,55]. As *Fusarium* fungi usually colonize the cereal grain from the external side [52], most of the mycotoxins strongly accumulate in the outer layers of grain and in products thereof, such as bran [55,56] and particularly grain dust [55]. Studies conducted on *Fusarium* infected wheat cultivars showed a higher fungal colonization and a major content of DON and ZEN in outer grain layer [43,52,56]. In addition, emerging *Fusarium* mycotoxins, such as enniatins, were found at a higher level in outer layers of wheat grain, such as bran [57,58].

In the present study, DON was found in a sufficient number of samples to set up a regression model. Linear regression analysis was performed using data from Table 3 (Figure 3, R^2^ DON = 0.68, *n* = 15). 

With respect to T-2 and HT-2, the contamination in the positive samples ranged from 6.8 to 54 µg/kg in grain (T-2 + HT-2 grain) and from 2.3 to 854 µg/kg in dust (T-2 + HT-2 dust) with an average ratio of T-2 + HT-2 dust/ T-2 + HT-2 grain ratio equal to 21 ± 19 µg/kg (*n* = 12 average ratio calculated on positive (>LOQ) samples of both wheat and corresponding dust). Due to the limited data available, it was not possible to further speculate on a possible correlation model.

The regression model implemented for DON was used to calculate the concentration in additional dust samples collected from trial 2 (sample set 2B). The dust and grain samples of sample set 2B were previously analyzed by the confirmatory method (LC-MS/MS). Therefore, the regression model was applied to the dust sample to calculate the DON concentration in the corresponding grain samples. Finally, the DON contamination in the whole grain obtained with the regression model (Figure 3) was compared with the concentration of DON directly detected in the whole grain lot by the confirmatory method (Table 3).

Overall, results matched quite well. For the first time, a linear model of DON contamination in the dust and the corresponding whole grain was verified at industrial level, confirming what was obtained in previous laboratory studies.

## 3. Discussion

The research carried out within the MycoKey project evaluates, for the first time, the applicability at industrial scale of the dust sampling approach to detect multiple toxins in grains. Results from the MycoKey trials, as well as the extensive review of currently available studies, allowed the identification of specific research needs to be addressed before evaluating the dust sampling, as an alternative or complementary approach to the official sampling. To date, neither a clear definition of grain dust for mycotoxin monitoring purpose nor any indication of the size of dust to be sampled was provided.

There are many different definitions for dust. According to the International Standardization Organization [59,60], “Dust” is defined as “small solid particles, conventionally taken as those particles below 75 µm in diameter, which settle out under their own weight, but which may remain suspended for some time”. The Glossary of Atmospheric Chemistry Terms” [61] defines dust as small, dry, solid particles projected into the air by natural forces, such as wind, volcanic eruption and by mechanical or man-made processes, such as crushing, grinding, milling, drilling, demolition, shoveling, conveying, screening, bagging and sweeping. Dust particles are usually in the size range from about 1 to 100 µm in diameter, and they settle slowly under the influence of gravity. In the sector of occupational safety and health, dust particles were classified in inhalable, thoracic and respirable fractions with a median aerodynamic diameter of 100, 10 and 4 µm, respectively. These fractions are expressed as curves which relate to the probability of inhalation or penetration to the thoracic or alveolar regions, as a function of the particle aerodynamic diameter. Each curve represents the sampling criterion to be achieved by any aerosol sampling instrument in order to measure the corresponding aerosol fraction [62]. The European standards EN 13205 [63] and EN 1540:2021 [64] specify the performance requirements for an aerosol sampler that include the assessment of the systematic deviation of the sampler, measurement uncertainty, measuring range, precision and impact of the main influential variables e.g., particle size, composition of particles, aerosol mass and variations in the sampling rate.

Regarding mycotoxin detection in grain by means of dust analysis, particles with several sizes were considered in previous studies (fine fraction < 50 µm [45,48,49], fraction of a size between 100 and 1000 µm [44], fraction < 1650 µm [50]). A good correlation (R^2^ > 0.68) was obtained for DON, regardless of the fractions collected. Furthermore, from the data reported in the literature for DON in wheat, the ratio of DON in dust and grain ranged from 5 to 13 and they are not dependent on the particle size of collected samples. Any speculation for the other mycotoxins and commodities was not possible, due to the lack of studies. However, in the context of grain monitoring, a dust definition including only fractions of particles of intermediate diameter (100–1000 µm) would be appropriate, since the fine particles (<50 µm) are ubiquitous and contain non-lot specific contaminants. These particles remain in the ambient air for a long time and settle on the following lots in elevators or storage facilities [44,65] and may easily adhere to the sampler’s surface, promoting carryover. From the research conducted, it appears that a standardization of the dust analysis approach is difficult to achieve, as the ratio of dust to grain varies for the same mycotoxin, even when collecting particles of the same size and the same commodity. Further studies are needed to assess whether the type of grain, the way both dust and grain are sampled, may influence the result. Moreover, the standardization of the dust sampling approach also implied the specification of the criteria that the sampling instrument must meet, such as sampling efficiency related to particles of a size between 100–1000 µm and other dust characteristics, such as composition, absorption capacity, solubility and hygroscopicity.

A more feasible solution than proposing a horizontal, standardized approach, would be to develop farm/company specific strategies to be adapted to different business needs. For instance, the dust sampling approach could be applied as a screening approach for self-monitoring and process management. In this context, a farm/company-specific correlation models should be implemented to establish a conversion factor between the value obtained with the dust sampler and the reference value of the grain lot. Once the calibration model was implemented, a cut-off value in dust (i.e., a threshold value in dust above or below which the grain sample is classified as non-compliant or compliant, respectively) can be estimated.

To set-up a meaningful calibration model, an adequate interval of the mycotoxins contamination level in grain samples is required (calibration range) as well as a uniform distribution of the contamination levels of mycotoxins in grain samples over the whole range. The calibration range depends on the aim of the sampling, specifically on the concentration of interest (f.i. for compliance testing, batches of grain with known amounts of mycotoxins encompassing the legal limits should be included in the calibration model). Moreover, at least three replicates for each calibration point should be collected in order to detect and remove any outliers from the model. As the experiments are very laborious, the dust sampling approach should be applied in parallel with the official sampling method (Commission Regulation EC No. 401/2006 and its amendment). At least three grain samples collected through the official protocol should be analyzed to estimate the precision of the result. It should be noted that correlation models, such as linear regression, are based on the assumption that the independent variable (mycotoxin concentration in grain sample) is not affected by error or that it is negligible. Therefore, the reliability of data is of utmost importance to build a meaningful model. Once a correlation model was implemented, it is then possible to derive a conversion factor, specific for a given mycotoxin and commodity.

The cut-off value should then be established in dust through validation experiments. To this aim, dust samples containing mycotoxins at the level of interest and negative dust samples (<LOD or equal to 0.2 of level of interest) should be analyzed. Although this procedure requires a considerable initial effort, it has several advantages: it allows a non-destructive and rapid identification of compliant good and, due to the high concentration of mycotoxins in dust samples, less sensitive techniques, such as rapid dipstick-based methods, could be applied for the analysis of dust samples, reducing analysis time and costs.

## 4. Conclusions

The study conducted within the MycoKey project, in line with previous works [44,46], shows that dust sampling and analysis represent a potential approach for non-destructive and rapid identification of compliant goods. Standardization of such procedure is however very difficult to implement. A more realistic solution would be to develop farm/company specific strategies that can be adapted to different business needs.

## 5. Materials and Methods

### 5.1. Reagents

Standard mycotoxins, NIV, DON, AFB1, T-2, HT-2 and ZEN were from Biopure Referenzen substanzen GmbH (Tulln, Austria), FB1, FB2, OTA was from Sigma-Aldrich (Milan, Italy) or from Cfm Oskar Tropitzsch (Marktredwitz, Germany). Acetonitrile and methanol (both high-performance liquid chromatography grade) and glacial acetic acid were purchased from VWR International (Milan, Italy), whereas the ammonium acetate was purchased from Sigma-Aldrich (Milan, Italy). Ultrapure water (18 MΩ) was produced by a Millipore Milli-Q system (Millipore, Bedford, MA, USA). The Oasis HLB prime column (3 cc, 60 mg) was purchased from Waters (Milan, Italy). Syringe filters Minisart RC4 (0.22 μm regenerated cellulose) were from Sartorius Stedim Biotech GmbH (Göttingen, Germany).

### 5.2. Dust Sampling Methodology

In the trial 1, 6 wagons of 95 m^3^ that were loaded with 60 t of wheat each, were sampled during unloading. Kernel samples were taken with a sampling probe according to EU Commission Regulation (EC) No 401/2006 [2]. In parallel, from the same outlet, dust was continuously collected throughout unloading with the rapidust^®^ mobile sampling system with a pressure release hole at the sampling probe. The hole was opened approximately once per minute to interrupt the air flow and hence avoid clogging of the sampling head. In the second trial (ii) within two days, 32 trucks delivering 25–30 t of wheat each, were controlled in their unloading. In this case, dust sampling was compared to standard intake control procedures that were applied in routine control at a grain processing facility. Dust sampling was performed throughout the unloading of the truck with the rapidust^®^ mobile sampling system with pressure release at the sampling probe. Kernel samples were taken using the pneumatic sample collector of the facility. Details on the sampling procedure for grain kernels and dust used in the two trials are reported in Table 4.

### 5.3. Analysis of Multi-Toxins by LC-MS/MS Method

#### 5.3.1. Mycotoxins Solutions

The following mixed mycotoxin solutions were prepared in acetonitrile, according to the concentrations specified in the following:Mixed stock solution A, to be used for wheat and maize grain spiking: NIV and DON 12.5 µg/mL; T-2 and HT-2 0.625 µg/mL; ZEN1.25 µg/mL; AFB1, 0.12 µg/mL;FB1 and FB2, 12.5 µg/mL and OTA, 0.105 µg/mL. This solution was diluted by 10 times to prepare calibrant solution for external matrix-matched calibration in grain samples;Mixed solution B, to be used for spiking experiments and to prepare the calibrant solution for the dust sample: NIV and DON 2.5 µg/mL; T-2 and HT-2 0.625 µg/mL; ZEN1.25 µg/mL; AFB1, 0.03 µg/mL; FB1 and FB2, 2.5 µg/mL and OTA, 0.04 µg/mL.

Calibrant solutions (five levels including blank) were prepared in blank sample extract solutions passing through an Oasis HLB column, according to the clean-up procedure described as follows. Appropriate volumes of the mixed standard solution were added to the column eluate before drying it down. Then the residue was redissolved with 1mL by adding first 300 μL of methanol, vortexing, and then adding 700 μL of water (to obtain a methanol/water ratio of 30/70, by vol.).

Matrix-matched calibrations were performed in the range 250–1500 µg/kg NIV, DON, FB1 and FB1; 12.5–75 T-2 and HT-2 µg/kg; 25–150 µg/kg ZEN and 2-12 µg/kg OTA for the grain samples and in the range 250–1500 µg/kg NIV, DON, FB1 and FB2; 2.62.5–375 µg/kg T-2 and HT-2; 125–750 µg/kg ZEN and 4-22 µg/kg OTA for dust samples.

Matrix matched calibrations were prepared for each matrix considered in this study (wheat and dust) using the relevant blank extract.

#### 5.3.2. Sample Preparation

Wheat and maize grain samples were finely ground by an ultra-centrifugal mill (ZM 200, Retsch GmbH, Retsch-Allee 1-5, 42781 Haan, Germany), equipped with a 500 mm sieve. Maize samples were used for validation purposes only (see Appendix A). Dust samples were not ground because of the small particles’ size (0.1–1 mm). Grain (2.5 g) and dust (0.5 g) samples were extracted first with methanol (10 mL for grain and 2 mL for dust) by 30-min shaking (extract A). After centrifugation (15 min, 4000× *g*), the extract A was removed, and the residue was extracted again with a mixture of acetonitrile/water (84/16) with 1% acetic acid (10 mL and 2 mL for grain and dust, respectively) by 30-min shaking (extract B). The extract B was recovered by centrifugation (15 min, 4000× *g*). Extracts A and B were unified and then diluted fourfold with water into a 7 mL glass vial. An aliquot of 2 mL (equivalent to 0.25 g of grain and dust sample) was diluted with 2 mL of acetonitrile. The extract was passed through the OASIS PRIME column, (3 cc, 60 mg) and 1 mL of water containing 0.1% of formic acid was added to recover all toxins. The eluate was dried under an air stream at 40 °C and reconstituted by adding first 300 μL of methanol, vortexing, and then adding 700 μL of water(to obtain a methanol/water ratio of 30/70, by vol.).Samples were filtered through a 0.22-µm RC syringe filter prior to injection into the LC-MS/MS apparatus.

For recovery experiments, individual sub-samples (2.5 g for wheat and maize grain and 0.5 g for dust) were spiked with an appropriate volume of the mixed mycotoxin solution A and B, respectively. Spiked samples were left overnight at room temperature to allow solvent evaporation and equilibration between analytes and matrix.

#### 5.3.3. LC-MS/MS Analysis

LC–MS/MS analyses were performed on a Waters AcquityUPLC I-class FTN system coupled to a Xevo TQ-S Triple Quadrupole mass Spectrometer (Waters, Milford, MA, USA) operating in positive electrospray ionization mode with the electrospray-ionization mode(ESI) source. The analytical column was an Acquity UPLC**^®^** HSS T3 column (100 mm × 2.1 mm I.D., 1.8 µm particle size). The column oven was set at 40 °C. The flow rate of the mobile phase was 0.4 mL/min and the injection volume was 10 µL. Eluent A was water and eluent B was methanol, both containing 0.5% acetic acid and 1mM ammonium acetate. For mycotoxin elution, the proportion of eluent B was kept constant at 2% for 2 min, then linearly increased to 50% in 4 min, then to increase to 80% over the next 2 min. Finally, it was raised to 98% and kept constant for 3 min. The column was re-equilibrated with 2% eluent B for 3.5 min. The parameters used for data acquisition in selected reaction monitoring (SRM) are reported in Appendix A. Examples of selected ion chromatograms of spiked and naturally contaminated dust samples are shown in Appendix A. MasslynxTM version 4.1 and Quanlynx**^®^** version 4.1 software (Waters, Milford, MA, USA) were used for data acquisition and processing.

#### 5.3.4. In House Verification of Method Performances

To evaluate recoveries, repeatability (RSD_r_) and within laboratory reproducibility (RSD_WLR_), the samples were spiked in triplicate at two different concentrations, namely 25% and 100% of the target level. In the case of wheat samples, the target levels for the validation experiments were set on the basis of the maximum permitted EU level, while for the dust samples, the expected value obtained from previous correlation studies performed by Eurofins were taken into account (Appendix A). The design was repeated on four different days (over a time period of 1 year). Estimated limits of quantification were 25 µg/kg for NIV, DON, FB1, FB2, 1.5 µg/kg for T-2, HT-2 and ZEN, 0.5 µg/kg for AFB1 and OTA. An estimate of the relative standard uncertainty (u′) associated with the results was calculated from the original Horwitz Equation (u′ = 2 ^1−0.5 log c^ for concentration 1.2 × 10^−7^ ≤ c ≤ 0.138) and the modified Horwitz Equation (u′= 0.22 c for concentration > 1.2 × 10^−7^). An estimate of the expanded uncertainty (U) corresponding to a confidence interval of approximately 95% was obtained by multiplying the relative standard uncertainty by a coverage factor of 2, U = 2u′ (Appendix A).

### 5.4. Calculation

The mycotoxin concentration in dust samples (x0) calculated by the model (y = b0 + bi y) and the corresponding 95% confidence interval (CI) were calculated according the following equation:x0 = (y − b0)/b1(1)
where b0 was the intercept of the linear regression model, b1 was the slope of the linear regression model, y was the concentration in dust samples (µg/kg):CI = 2 × sx0(2)
where sx0 was the standard deviation of the forecasted concentration x0 from y0 through the regression line.
(3)$$\mathrm{sx}0=\frac{\mathrm{syx}}{b1}\sqrt{1/r+1/n+{\left(y0-\mathrm{ym}\right)}^{2}/(b_{1}^{2}\;*\;\Sigma {\left(\mathrm{xi}-\mathrm{xm}\right)}^{2}})$$
y0: mycotoxin concentration in dust, b1 slope, *r*: replicates of calibration points; *n* calibration data points in linear regression, xm and ym were the average of x and y data point of linear regression, respectively, xi is the x values of each data point of the linear regression.

## Figures and Tables

**Figure 1 toxins-14-00381-f001:**
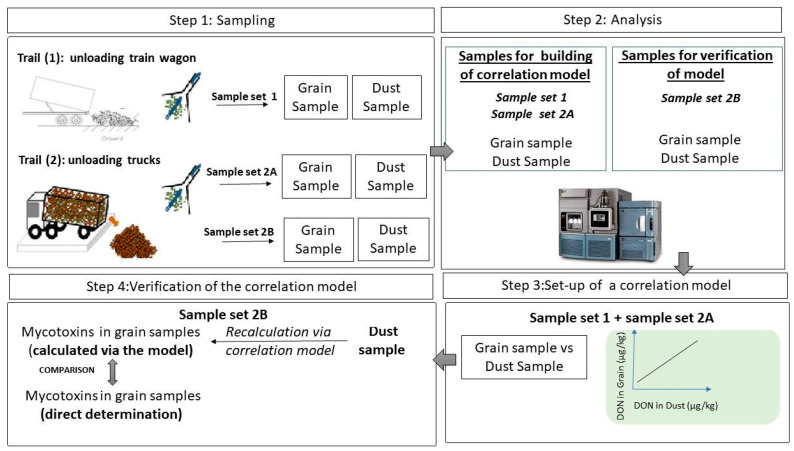
Dust sampling approach and relevant steps: 1. sampling of dust and grain applied to the two trials in the EU industrial site, specifically the control of wheat unloading from train wagons (trial 1) and the control of wheat unloading from trucks (trial 2, where two different sample sets were collected, namely sample set 2A and sample set 2B); 2. analysis by a confirmatory method (LC-MS/MS); 3. set-up of a correlation model using the sample set 1 (from trial 1) and sample set 2A (from trial 2); 4. verification of the correlation model, comparing the mycotoxins contamination in grain samples of the sample set 2B, calculated by the implemented model and the mycotoxin contamination in the grain sample directly measured by LC-MS/MS.

**Figure 2 toxins-14-00381-f002:**
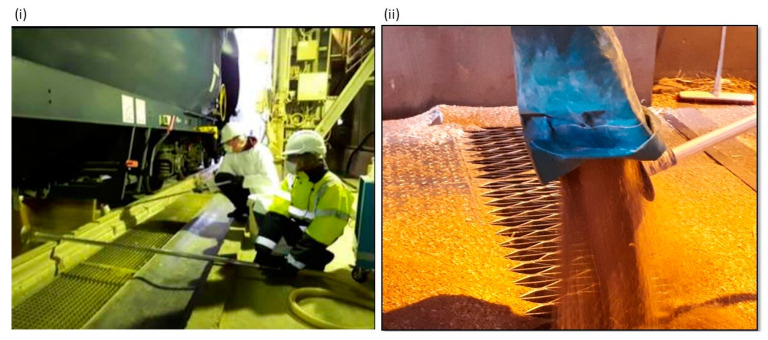
(**i**) Control of wheat unloading from train wagons; (**ii**): Control of wheat unloading from trucks at a grain processing facility.

**Figure 3 toxins-14-00381-f003:**
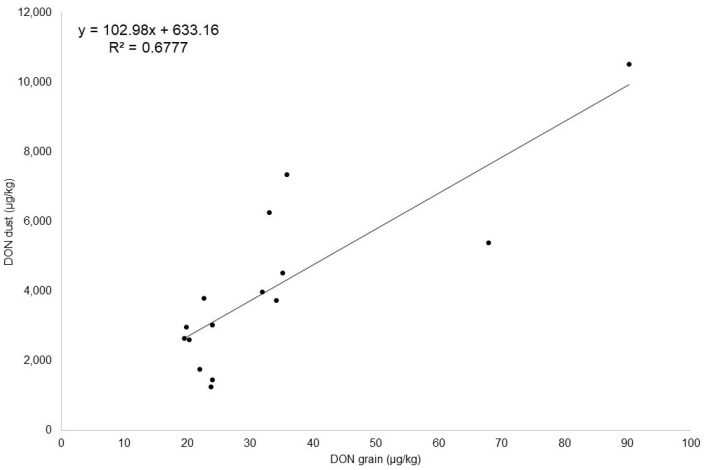
Regression line showing the correlation of deoxynivalenol (DON) contaminations in wheat grain and in wheat dust.

**Table 1 toxins-14-00381-t001:** Distribution studies available for each mycotoxin in several commodities. The symbol (●) represents one distribution study for the relative mycotoxin/commodity.

		Aflatoxins	FBs	DON	OTA
**Cereals**	**Barley**			**●**	
	**Maize**	**●●●●**	**●●●**	**●**	**●**
	**Oats**				**●**
	**Wheat**			**●**	**●**
**Nuts**	**Almonds**	**●●●**			
	**Pistachios**	**●**			
	**Brazil Nuts**	**●**			
	**Hazelnuts**	**●**			
	**Peanuts**	**●●●●●●●●**			
**Ginger**		**●●**			**●●**
**Dried Figs**		**●**			
**Coffee Beans**					**●●●**
**Cotton Seed**		**●●●●**			

FBs: fumonisin B1 and fumonisin B2; OTA: ochratoxin A; DON: deoxynivalenol.

**Table 2 toxins-14-00381-t002:** Results of the LC-MS/MS multi-mycotoxin analysis in samples from trial 1 (=control of wheat unloading from train wagons) and trial 2 (=control of wheat unloading from truck loads). AFB1, FB1 and FB2 were not detected (<LOQ) in any of the samples. (−) means < LOQ. LOQs were 20 µg/kg for NIV, DON, FB1, FB2, 10 µg/kg for ZEN, 1.5 µg/kg for T2, HT2, 0.5 µg/kg for AFB1, OTA. In the trial 2, the label A and B indicated the two dust samples collected from the same truck for the sample set 2A and the sample set 2B, respectively.

					µg/kg					
	NIV		DON		T2+HT2		ZEN		OTA	
	Grain	Dust	Grain	Dust	Grain	Dust	Grain	Dust	Grain	Dust
**Trial 1**										
**Sample set 1**										
Wagon 1	(−)	308	(−)	1097	(−)	367	(−)	(−)	(−)	(−)
Wagon 2	(−)	176	(−)	1057	(−)	296	(−)	(−)	(−)	(−)
Wagon 3	(−)	159	(−)	551	(−)	299	(−)	(−)	(−)	(−)
Wagon 4	(−)	95	(−)	490	(−)	196	(−)	(−)	(−)	(−)
Wagon 5	(−)	256	(−)	860	(−)	660	(−)	(−)	(−)	(−)
Wagon 6	(−)	161	(−)	686	(−)	368	(−)	(−)	(−)	(−)
**Trial 2**										
**Sample set 2A**										
Truck 1	(−)	85	(−)	2018	(−)	20	(−)	51	(−)	(−)
Truck 2	(−)	114	(−)	795	(−)	8.7	(−)	(−)	(−)	3.2
Truck 3	(−)	77	20	2648	(−)	74	(−)	(−)	(−)	(−)
Truck 4	(−)	79	34	3745	(−)	12	(−)	308	(−)	53
Truck 5 A	(−)	42	24	1260	(−)	15	(−)	(−)	(−)	(−)
Truck 6 A	(−)	162	35	4539	(−)	2.3	(−)	(−)	(−)	0.4
Truck 7 A	(−)	344	(−)	1129	(−)	19	(−)	(−)	(−)	(−)
Truck 8	(−)	204	(−)	1095	52	391	(−)	(−)	(−)	(−)
Truck 9	(−)	58	(−)	1228	(−)	14	(−)	(−)	(−)	(−)
Truck 10	(−)	14	(−)	353	(−)	4.5	(−)	(−)	(−)	6.3
Truck 11 A	(−)	255	(−)	1194	9.5	317	(−)	(−)	(−)	(−)
Truck 12	(−)	(−)	(−)	4206	(−)	5.9	(−)	(−)	(−)	(−)
Truck 13	(−)	92	(−)	1163	54	363	(−)	(−)	(−)	(−)
Truck 14	(−)	173	23	3803	(−)	4.9	(−)	(−)	(−)	(−)
Truck 15 A	(−)	94	(−)	2174	(−)	8.9	(−)	(−)	(−)	(−)
Truck 16 A	(−)	183	36	7347	(−)	14	(−)	(−)	(−)	(−)
Truck 17	(−)	248	(−)	5094	8.5	190	(−)	79	(−)	6.2
Truck 18	(−)	358	(−)	4367	(−)	446	(−)	(−)	(−)	(−)
Truck 19 A	(−)	172	33	6267	(−)	88	(−)	277	(−)	14
Truck 20	(−)	696	(−)	1628	6.8	191	(−)	(−)	(−)	1.5
Truck 21	(−)	335	20	2972	31	363	(−)	(−)	(−)	(−)
Truck 22	(−)	190	(−)	3575	(−)	194	(−)	(−)	(−)	2.0
Truck 23	(−)	168	68	5404	(−)	12	(−)	(−)	(−)	(−)
Truck 24	(−)	430	20	2606	(−)	16	(−)	(−)	(−)	0.2
Truck 25	(−)	398	24	3028	26	347	(−)	(−)	(−)	(−)
Truck 26	(−)	208	(−)	2250	(−)	359	(−)	(−)	(−)	(−)
Truck 27	(−)	264	(−)	1206	12	854	(−)	(−)	(−)	(−)
Truck 28 A	(−)	249	(−)	1268	19	605	(−)	(−)	(−)	1.1
Truck 29	(−)	331	24	1463	27	35	(−)	232	(−)	0.8
Truck 30	(−)	187	22	1760	47	388	(−)	(−)	(−)	(−)
Truck 31	(−)	297	90	10,531	12	154	0 (−)	(−)	(−)	3.3
Truck 32	(−)	281	32	3977	(−)	(−)	(−)	(−)	(−)	(−)
**Sample set 2B**										
Truck 5 B	(−)	163	24	1161	(−)	72	(−)	(−)	(−)	(−)
Truck 6 B	(−)	185	35	5168	(−)	2.4	(−)	(−)	(−)	(−)
Truck 7 B	(−)	560	(−)	2631	(−)	199	(−)	(−)	(−)	(−)
Truck 11 B	(−)	291	(−)	932	9.5	305	(−)	(−)	(−)	(−)
Truck 15 B	(−)	103	(−)	2638	(−)	12	(−)	(−)	(−)	(−)
Truck 16 B	(−)	237	36	7282	(−)	16	(−)	(−)	(−)	(−)
Truck 19 B	(−)	163	33	6572	(−)	138	(−)	(−)	(−)	(−)
Truck 28 B	(−)	154	(−)	841	19	447	(−)	(−)	(−)	(−)

**Table 3 toxins-14-00381-t003:** Results for DON in wheat truck loads: results were obtained from laboratory analyses by HPLC-MS/MS from ground grain samples and from dust samples by the correlation model (regression line in Figure 3). * DON concentration calculated by HPLC-MS/MS method in dust samples and grain samples and the corresponding expanded uncertainty (at 95% confidence level) calculated by Horwitz equation ** The DON concentration calculated by the model (x0) and the corresponding 95% confidence interval (CI)was reported for each sample *.

Sample Name	DON Contamination in Dust Samples * × 10^3^ (µg/kg)	DON Contamination in Wheat Grains × 10^2^ (µg/kg)	
		Calculated by the Correlation Model **	Direct Determination in Grain Sample *
Sample set 2B			
Truck 5 B	1.2 ± 0.4	0.1 ± 0.4	0.2 ± 0.1
Truck 6 B	5.2 ± 1.3	0.4 ± 0.4	0.4 ± 0.2
Truck 7 B	2.6 ± 0.7	0.2 ± 0.4	(−)
Truck 11 B	0.9 ± 0.3	0.0 ± 0.4	(−)
Truck 15 B	2.6 ± 0.7	0.2 ± 0.4	(−)
Truck 16 B	7.3 ± 1.7	0.6 ± 0.4	0.4 ± 0.2
Truck 19 B	6.6 ± 1.5	0.6 ± 0.4	0.3 ± 0.1
Truck 28 B	0.8± 0.3	0.0 ± 0.4	(−)

**Table 4 toxins-14-00381-t004:** Features of the sampling method applied in the two trials. Fraction weight, type of sampling (dynamic or static), number of incremental samples of sampled fractions, aggregate sample weight, sampling interval (unloading time (sec)/number of increment) and duration of sampling procedure (time required to obtain one aggregate sample from each wagon/truck) are specified.

	Trial (i): Unloading of Train Wagon		Trial (ii): Unloading of Wheat Trucks	
	Regulation EC 401/2006	Dust Sampling	Standard Intake Control Procedures	Dust Sampling
Fraction Weight (Tons)	60	60	25–30	25–30
Type of Sampling	Dynamic	Dynamic	Static	Dynamic
Number of Incremental Sample	100	1	3	1
Aggregate Sample Weight (kg)	10	0.005 *	3	0.005 *
Sampling interval (sec)	9	n.a.	n.a.	n.a.
Duration of Sampling Procedure (min)	15–20	15–20	10	10

n.a. = not applicable. * The size of the collected dust samples was between 0.005 and 0.01 kg.

## Data Availability

The data presented in this study are available in this article and Appendix A.

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
