# Peer review of "Mycotoxin Analysis of Grain via Dust Sampling: Review, Recent Advances and the Way Forward: The Contribution of the MycoKey Project"

_toxins, 2022, doi:10.3390/toxins14060381_

Round 1

Reviewer 1 Report

The article entitled "Mycotoxin analysis of grain via dust sampling: review, recent advances and the way forward. The contribution of the Myco-Key Project" presents an interesting review of sampling protocols for the official control of mycotoxin levels in food. It shows that more work is needed to implement sampling plans with the capacity to support official control. It describes the research carried out in the framework of the European Horizon 2020 project MycoKey aimed at solving this problem.

However, the results of this project show that standardisation of this sampling procedure is very difficult to achieve and that further work is needed in this direction. I consider that the work provides interesting information for researchers in this field and therefore I consider the work to be publishable in its present form,

Author Response

Dear Reviewer, 
Thank you for your comments

Reviewer 2 Report

The paper entitled: Mycotoxin analysis of grain via dust sampling: review, recent 2 advances and the way forward. The contribution of the Myco-3 Key Project, deals with the sampling protocols for the official control of the levels of mycotoxins in foodstuffs in the environment of the European Horizon 2020 MycoKey project devoted to evaluate the applicability at industrial scale of the dust sampling approach to detect multiple mycotoxins in grains. The paper describes two trials in an EU industrial site and concludes that dust sampling and mycotoxin analysis represent a fitness for purpose approach for non-destructive and rapid identification of wheat commodities compliant to maximum permitted levels. The paper is interesting for those researchers working in the field of mycotoxin sampling protocols. Albeit the paper does not implement new methodology, but uses established methodology, the large number of samples studied and the in field application of results makes the paper suitable for publication as it is.

Author Response

(The authors gave the same response as above.)

Reviewer 3 Report

very well-written draft. the aim of study is highly innovative.

well presented results. it can be published as-is.

I have comment on additional correction: representative LC/MS chromatogram of dust sample can be presented, eg. in supl. mat. section.

Author Response

Dear Reviewer, 
Thank you for your comments.  According to your suggestion, representative LC/MS chromatograms of naturally contaminated dust and spiked dust extract were added in the supplementary section. All the changes are marked in red color